# Unsupervised Disentanglement of Pose, Appearance and Background from Images and Videos

## Abstract

Unsupervised landmark learning is the task of learning semantic keypoint-like representations without the use of expensive input keypoint-level annotations. A popular approach is to factorize an image into a pose and appearance data stream, then to reconstruct the image from the factorized components. The pose representation should capture a set of consistent and tightly localized landmarks in order to facilitate reconstruction of the input image. Ultimately, we wish for our learned landmarks to focus on the foreground object of interest. However, the reconstruction task of the *entire* image forces the model to allocate landmarks to model the background. This work explores the effects of factorizing the reconstruction task into separate foreground and background reconstructions, conditioning only the foreground reconstruction on the unsupervised landmarks. Our experiments demonstrate that the proposed factorization results in landmarks that are focused on the foreground object of interest. Furthermore, the rendered background quality is also improved, as the background rendering pipeline no longer requires the ill-suited landmarks to model its pose and appearance. We demonstrate this improvement in the context of the video-prediction task.

## 1 Introduction

Pose prediction is a classical computer vision task that involves inferring the location and configuration of deformable objects within an image. It has applications in human activity classification, finding semantic correspondences across multiple object instances, and robot planning to name a few. One of the caveats of this task is that annotation is very expensive. Individual object "parts" need to be carefully and consistently annotated with pixel-level precision. Our work focuses on the task of *unsupervised* landmark learning, which aims to find unsupervised pose representations from image data without the need for direct pose-level annotation.

A good visual landmark should be tightly localized, consistent across multiple object instances, and grounded on the foreground object of interest. Tight localization is important because many objects (such as persons) are highly deformable. A landmark localized to a smaller, rigid area of the object will offer more precise pose information in the event of object motion. Consistency across multiple object instances is also important, as we wish for our landmarks to apply to all instances within a visual category. Finally, and most relevant to our proposed method, we want our landmarks to focus on the foreground objects. A landmark that fires on the background is a wasted landmark, as the background is constantly changing, and yields little information regarding the pose of our foreground object of interest.

Many unsupervised landmark learning methods perturb an input training image with various transformations, then require the model to learn semantic correspondences across the transformed variants to piece together the unaltered input image. The primary issue with this approach is it penalizes the *entire* image reconstruction when we care only about the foreground, resulting in landmarks being allocated to the background. This poses a number of issues, including increased memory requirements (more landmarks required to capture the foreground) and lower landmark reliability (landmarks assigned to background are unstable). Our proposed method aims to reduce the likeli-

hood of landmarks being allocated to the background, thereby improving overall landmark quality and reducing the number of landmarks required to achieve state-of-the-art performance.

Our work builds upon existing methods in image-reconstruction-guided landmark learning techniques (Jakab et al., 2018; Lorenz et al., 2019). We explicitly encourage our model to factorize the reconstruction task into separate foreground and background reconstructions, where only the foreground reconstruction is conditioned on learned landmarks. Our contributions are as follows:

1. We propose an improvement to reconstruction-guided unsupervised landmark learning that allows the landmarks to better focus on the foreground.
2. We demonstrate through empirical analysis that our proposed factorization allows for state-of-the-art landmark results with fewer learned landmarks, and that fewer landmarks are allocated to modeling background content.
3. We demonstrate that the overall quality of the reconstructed frame is improved via the factorized rendering, and include an application to the video-prediction task.

## 2    RELATED WORKS

Our work builds upon prior methods in unsupervised discovery of image correspondences (Thewlis et al., 2017b; Zhang et al., 2018b; Suwajanakorn et al., 2018; Thewlis et al., 2017a; Kanazawa et al., 2016; Jakab et al., 2018; Lorenz et al., 2019). Most relevant here are Jakab et al. (2018) and Lorenz et al. (2019), which learn the latent landmark activation maps via an image factorization and reconstruction pipeline. Each image is factored into pose and appearance representations and a decoder is trained to reconstruct the image from these latent factors. The loss is designed such that accurate image reconstruction can only be achieved when the landmarks activate at consistent locations between an image its TPS-warped variant. Lorenz et al. (2019) specifically improves upon the method proposed by Jakab et al. (2018) such that instead of representing the appearance information as a single vector for the entire image, there is a separate appearance encoding for each landmark in the pose representation. One limitation of these works is that the appearance and pose vectors also need to encode background information in order to reconstruct the entire image. Our work attempts to resolve this limitation by introducing unsupervised foreground-background separation into the pipeline, using the pose and appearance vectors for only the foreground rendering.

There are few other works that propose to separate foreground and background in image rendering tasks. Balakrishnan et al. (2018) separates foreground and background for image synthesis in an unseen pose, but their method relies on supervised 2D keypoints. Rhodin et al. (2018) and Rhodin et al. (2019) separate background from foreground for single and multi-person pose-estimation. In both works, the background images are computed by taking the median pixel value across all frames, and therefore require video sequence data with perfectly static backgrounds. Instead, our approach trains a network to synthesize a clean background from any input frame. It is therefore more forgiving with respect to background variation, and can even handle thin-plate-spline warped backgrounds after overfitting to the training data. This allows us to use our method on non-video datasets such as CelebA faces (Liu et al., 2015).

## 3    METHOD

Our method extends the pipeline proposed by Lorenz et al. (2019). At a high-level, it reconstructs an image from two perturbed variants: one where the appearance (color, lighting, texture) information is perturbed, and one where the pose (position, orientation) of the object is perturbed. The model must learn to extract the pose information from the appearance-perturbed image, and appearance information from the pose-perturbed image. The model will learn a set of landmarks in the process as a means to spatially-align the information extracted from the two sources in order to reconstruct the original image.

Our extension to Lorenz et al. (2019) factorizes the final reconstruction into separate foreground and background renders, where only the foreground is rendered conditioned on the landmark positions. The background will be inferred directly from the pose-perturbed input image with a simple UNet (Ronneberger et al., 2015). We want our UNet to have a limited capacity for handling complex changes in pose. The remaining complex pose changes (e.g. limb motion, object rotations) will

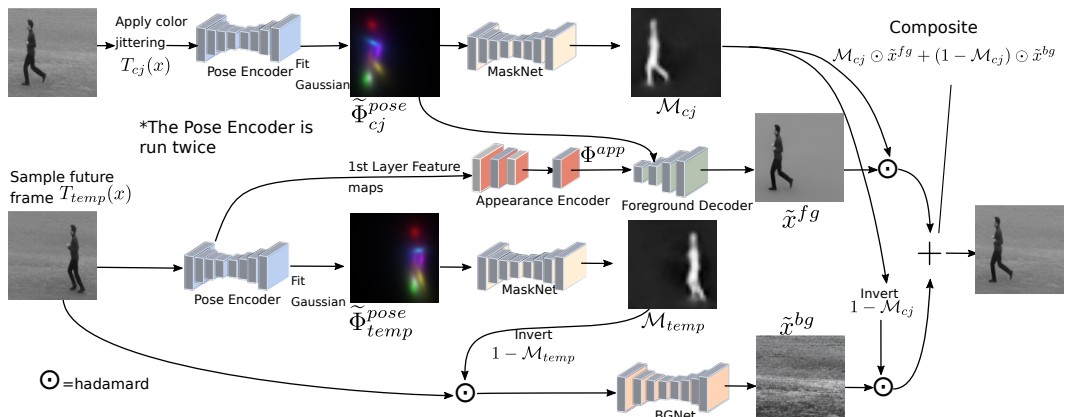

Figure 1: This figure depicts an overview of our training pipeline on video data. Given an image frame $x$, we produce $T_{cj}(x)$ and $T_{temp}(x)$, which are appearance and pose perturbed variants of $x$ respectively. The model learns to combine the appearance information from $T_{temp}(x)$, and combine it with the pose from the $T_{cj}(x)$ in order to reconstruct foreground object from the foreground decoder. Foreground masks are predicted as part of the pipeline to separate the foreground rendering from the background rendering. Specifically, the background is rendered from a UNet that learns to extract clean backgrounds from $T_{temp}(x)$. This allows the learned pose representation to focus on the more dynamic foreground object. The pose encoder and MaskNet are each depicted twice as they are applied twice during the forward pass.

then be captured by the more flexible landmark representations. During training, *this factorization is guided by a translating foreground/static background assumption*, though we demonstrate that landmark quality is improved even when this assumption is held weakly.

### 3.1 MODEL COMPONENTS

Our full pipeline comprises five components: the pose and appearance encoders, foreground decoder, background reconstruction subnet, and foreground mask subnet.

The goal of the *pose encoder* $\Phi^{pose} = Enc^{pose}(x)$ is to take an input image $x$ and output a set of unsupervised part activation maps. Critically, we want these part activation maps to be invariant to changes in local appearance, as well as to be consistent across deformations. A heatmap that activates on a person's right hand should be invariant across varying skin tones and lighting conditions, as well track the right hand's location across varying deformations and translations.

The *appearance encoder* $\Phi^{app} = Enc^{app}(x; Enc^{pose}(x))$ extracts local appearance information, conditioned on the pose-encoder's activation maps. Given an input image $x$, the pose encoder will first provide $K \times H \times W$ part activation maps $\Phi^{pose}$. To extract local appearance vectors, the appearance encoder projects the image to a $C \times H \times W$ appearance feature map $M^{app}$. We compute the appearance vector for the $k$th pose activation map as:

$$\Phi^{app}_{k,c} = \sum_i^H \sum_j^W \Phi^{pose}_{k,i,j} M^{app}_{c,i,j} \text{ for } c = 1...C, \tag{1}$$

giving us $K$ $C$-dimensional appearance vectors. Here, each activation map in $\Phi^{pose}$ is softmax-normalized.

The method pipeline attempts to reconstruct the original input image by combining the pose information from the $K$ activation maps with the pooled appearance vectors for each of the $K$ parts. As in Lorenz et al. (2019), we fit a 2D Gaussian to each activation of the $K$ activation maps by estimating their respective means and either estimating or using a pre-determined covariance. Each part is represented by $\widetilde{\Phi}^{pose}_k = (\mu_k, \Sigma_k)$, where $\mu_k \in \mathbb{R}^2$ and $\Sigma_k \in \mathbb{R}^{2 \times 2}$. The 2D Gaussian approximation forces each part activation map into a unimodal representation with a simple parameterization, thereby enforcing that each landmark appears in at most one location per image.

The *foreground decoder* ($FGDec$) and *background reconstruction subnet* ($BGNet$) are networks that attempt to reconstruct the foreground and background respectively. Our foreground decoder is based on the architecture proposed in SPADE (Park et al., 2019). In SPADE, semantic maps are used to predict spatially-aware affine transformation parameters for normalization schemes such as InstanceNorm. Herein, we project the 2D Gaussian parameters from $\widetilde{\Phi}^{pose}$ to a heatmap of the target output width and height to use as semantic maps in the SPADE architecture. Following Lorenz et al. (2019), we use the formula:

$$s(k,l) = \frac{1}{1 + (l - \mu_k)^\mathsf{T} \Sigma_k^{-1} (l - \mu_k)} \tag{2}$$

where $s(k,l)$ is the heatmap value for part map $k$ at coordinate location $l$. In addition to feeding $s(k,l)$ as a semantic map to SPADE, individual appeerence vectors are also projected onto their respective heatmap to create a localized appearance encoding to be fed into the decoder. Please see section 3.4 of Lorenz et al. (2019) for details on this projection.

Unlike the foreground decoder, which is conditioned on bottlenecked pose-appearance representation, the $BGNet$ is given direct access to image data, albeit the pose-perturbed variant of the input. Given a static background video sequence, we assume it is easier for the $BGNet$ to learn to directly copy background pixels (and remove the foreground when necessary) than it is for the pose-appearance factorization to learn to model the background. In the absence of a $BGNet$-like module, several landmarks will be allocated to capture the "pose" of the background, despite being ill-suited for such a task.

The final module is the *foreground mask subnet* ($MaskNet$), which infers the blending mask to composite the foreground and background renders. It can be interpreted as a foreground segmentation mask and is conditioned on $\widetilde{\Phi}^{pose}$.

## 3.2 Training Pipeline

All network modules are jointly trained in a fully self-supervised fashion, using the final image reconstruction task as guidance. We follow the training method as detailed in Lorenz et al. (2019), with the addition of our proposed factorized rendering pipeline in the reconstruction phase. An illustration of this pipeline is depicted in Fig. 1.

Training involves reconstructing an image from its appearance and pose perturbed variants, learning to extract the un-perturbed element from each variant. As with Lorenz et al. (2019), we use color jittering to construct the appearance-perturbed variant $T_{cj}(x)$. When training from video data, we temporally sample a frame 3 to 60 timesteps apart from the same scene to attain the pose-perturbed variant $T_{temp}(x)$. However, in the absence of video data, we use thin-plate-spline warping to perturb pose $T_{tps}(x)$. In general, our method is able to work with both $T_{temp}(x)$ and $T_{tps}(x)$, though TPS-warping has the downside of also warping the background pixels, making the task of $BGNet$ more difficult. Let $\widetilde{\Phi}^{pose}$ be the gaussian-heatmap fitted to the raw activation map $\Phi^{pose}$, and let $\odot$ represent element-wise multiplication. Our training procedure can be expressed as follows:

$$\Phi_{cj}^{pose} = Enc^{pose}(T_{cj}(x)) \text{ and } \Phi_{temp}^{pose} = Enc^{pose}(T_{temp}(x)) \tag{3}$$

$$\Phi^{app} = Enc^{app}(T_{temp}(x); Enc^{pose}(T_{temp}(x))) \tag{4}$$

$$\mathcal{M}_{cj} = MaskNet(\widetilde{\Phi}_{cj}^{pose}) \text{ and } \mathcal{M}_{temp} = MaskNet(\widetilde{\Phi}_{temp}^{pose}) \tag{5}$$

$$\tilde{x}^{fg} = FGDec(\widetilde{\Phi}_{cj}^{pose}, \Phi^{app}) \text{ and } \tilde{x}^{bg} = BGNet((1 - \mathcal{M}_{temp}) \odot T_{temp}(x)), \tag{6}$$

$$\tilde{x} = \mathcal{M}_{cj} \odot \tilde{x}^{fg} + (1 - \mathcal{M}_{cj}) \odot \tilde{x}^{bg} \tag{7}$$

where the goal is to minimize the reconstruction loss between the original input $x$ and the reconstruction $\tilde{x}$. As can be seen, neither the shape encoder nor the appearance encoder are ever given direct access to the original image $x$. The pose information feeding into the foreground decoder $FGDec(\cdot, \cdot)$ is based on the color-jittered input image, where only the local appearance information is perturbed. The appearance information is captured from $T_{temp}(x)$ (or $T_{tps}(x)$), where the pose information is perturbed. Notice the shape encoder is also executed on both the pose-perturbed and color-jittered input images. This is necessary to map the localized appearance information for a particular landmark from its location in the pose-perturbed image to its unaltered position in $T_{cj}(x)$.

Table 1: Evaluation of landmark accuracy on Human3.6M 1a and BBC Pose 1b. Human3.6M error is normalized by image dimensions. For BBC Pose, we report the percentage of annotated keypoints predicted within a 6-pixel radius of the ground truth.

| Human3.6M | | Error |
|---|---|---|
| supervised | Newell et al. (2016) | 2.16 |
| unsup. | Thewlis et al. (2017b) | 7.51 |
| | Zhang et al. (2018a) | 4.91 |
| | Lorenz et al. (2019) | 2.79 |
| | Baseline (temp) | 3.07 |
| | Baseline (temp,tps) | 2.86 |
| | Ours | 2.73 |

(a)

| BBC Pose | | Acc. |
|---|---|---|
| supervised | Charles et al. (2013) | 79.9% |
| | Pfister et al. (2015) | 88.0% |
| unsup. | Jakab et al. (2018) | 68.4% |
| | Lorenz et al. (2019) | 74.5% |
| | Baseline (temp) | 73.3% |
| | Baseline (temp, tps) | 73.4% |
| | Ours | 78.8% |

(b)

Finally, the predicted foreground-background masks are computed for both the appearance and pose perturbed variants: $\mathcal{M}_{cj}$ and $\mathcal{M}_{temp}$ respectively. $\mathcal{M}_{cj}$ should have a foreground mask corresponding to the original foreground's pose, and is used to blend the foreground and background renders in the final step. $\mathcal{M}_{temp}$ is the foreground mask for the pose-perturbed input image, and assists the $BGNet$ in removing foreground information from its background render. Refer to Appendix A for architecture, loss, and training parameters.

## 4 EXPERIMENTS

Here, we analyze the effect of introducing foreground-background separation into an unsupervised-landmark pipeline. Through empirical analysis, we demonstrate that the learned landmarks less used for capturing background information, thereby improving overall landmark quality. Landmark quality is evaluated by using linear regression to map the unsupervised landmarks to annotated keypoints, with the assumption that well-placed, spatially consistent landmarks lead to low regression error. Finally, we include an additional application of our method in the video prediction task, demonstrating how the factorized rendering pipeline improves the overall rendered result.

### 4.1 DATASETS

We evaluate our method on Human3.6M (Ionescu et al., 2013), BBC Pose (Charles et al., 2013), CelebA (Liu et al., 2015), and KTH (Schuldt et al., 2004). Human3.6M is a video dataset that features human activities recorded with stationary cameras from multiple viewpoints. BBC Pose dataset contains video sequences featuring 9 unique sign language interpreters. Individual frames are annotated with keypoint annotations for the signer. While most of the motion is from the hand gestures of the signers, the background features a constantly changing display that makes clean background separation more difficult. CelebA is an image-only dataset that features keypoint-annotated celebrity faces. As with prior works, we separate out the smaller MAFL subset of the dataset, train our landmark representation on the remaining CelebA training set, and perform the annotated regression task on the MAFL subset. The KTH dataset comprises videos of people performing one of six actions (walking, running, jogging, boxing, handwaving, hand-clapping). We use KTH for our video prediction application. Additional preprocessing details are given in Appendix A.

### 4.2 UNSUPERVISED LANDMARK EVALUATION

As with prior works (Jakab et al., 2018; Thewlis et al., 2017a), we fit a linear regressor (without intercept) to our learned landmark locations from our pose representation to supervised keypoint coordinates. Following Jakab et al. (2018), we create a loose crop around the foreground object using the provided keypoint annotations, and evaluate our landmark learning method within said crop. Importantly, most prior methods have not released their evaluation code for all datasets, thus we were not able to control for cropping parameters and coordinate space. The former affects the relative size and aspect ratio of the foreground object to the input frame, whereas the latter affects the regression results in the absence of a bias term. As such, external comparisons on this task should be interpreted as a rough comparison at best, and that the reader focus on the comparison against our internal baseline, which is our rough implementation of Lorenz et al. (2019). We include our cropping details in Appendix A.

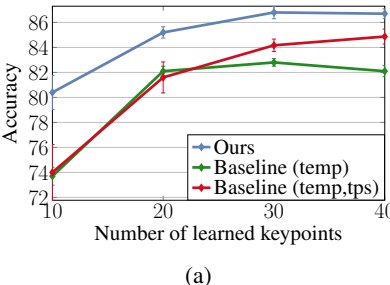 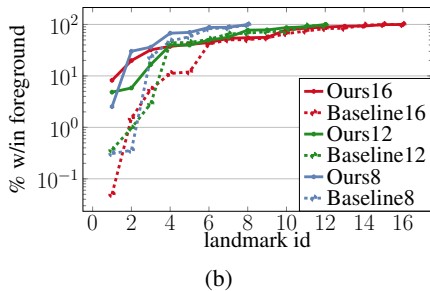

(a)                                      (b)

Figure 2: Landmark analysis experiments. 2a plots the BBC validation dataset keypoint accuracy versus number of learned keypoints. By factorizing out the background rendering, we are able to achieve better landmark-to-annotation mappings with fewer landmarks than the baseline. 2b plots the percentage of the per-landmark normalized activation maps contained within the provided foreground segmentation masks on Human3.6M, sorted in ascending order. We compare our model against our baseline at 8, 12, and 16 learned landmarks. We see that the least-contained landmarks in the proposed approach are significantly more contained than those of the baseline.

We report our regression accuracies on Human3.6M, BBC, and CelebA/MAFL, with the first two being video-based datasets and the last being image only. Results are shown in Tables 1a, 1b, and 2 respectively. For the video datasets, we found it best to use only $T_{temp}(x)$ to sample perturbed poses from future frames during training. Only $T_{tps}(x)$ was possible for CelebA/MAFL. Our primary baseline is our model without the explicit foreground-background separation. For this baseline, we report results using $T_{temp}$-only (Baseline (temp)) as well as both $T_{temp}$ and $T_{tps}$ (Baseline (temp,tps)). In all cases, we demonstrate that including factorized foreground-background rendering improves landmark quality compared to the controlled baseline model. We also believe our performance is competitive if not state-of-the-art based on our best-attempt at matching cropping and regression protocols for external comparisons. The results on CelebA demonstrate that our method works even given very weak static background assumptions. This is because $T_{tps}(x)$ indiscriminately warps the entire image, creating a pose-perturbed variant with a heavily deformed background. Further discussion in Appendix B.

Next, we analyze how factorizing out the background rendering influences landmark quality. In Fig. 2a, we present an ablation study where we measure the regression-to-annotation accuracy against the number of learned landmarks. Compared to our baseline models, we can see that the background-factorization allows us to achieve better accuracy with fewer landmarks, and that the degradation is less steep. Further, in Table 2, we include a No Mask baseline which is our proposed model but sans predicted blending masks. Here, we combine foreground and background directly with: $\tilde{x} = \tilde{x}^{fg} + \tilde{x}^{bg}$. This variant also improves over the unfactorized baseline, though the full pipeline still performs best.

Table 2: Landmark evaluation on MAFL using 10 landmarks. Prediction error is scaled by inter-ocular distance. While we can only use $T_{tps}$ to sample pose perturbations on this non-video dataset, we still see strong improvements over the baseline. We also ablate the use of the masks $\mathcal{M}$ in Ours (No Mask), where no masks are predicted and the predicted $\tilde{x}^{fg}$ and $\tilde{x}^{bg}$ are directly elementwise-added.

| MAFL | Error |
|---|---|
| Thewlis et al. (2017b) | 6.32 |
| Zhang et al. (2018a) | 3.46 |
| Lorenz et al. (2019) | 3.24 |
| Jakab et al. (2018) | 3.19 |
| Baseline (tps) | 4.34 |
| Ours (No Mask) | 2.88 |
| Ours | 2.76 |

One of our primary claims is that by factorizing foreground and background rendering in the training pipeline, we allow the landmarks to focus on modeling the pose and appearance of the foreground objects, leaving the background rendering task to a less expressive, but easier to learn mechanism. We attempt to validate this claim on the Human3.6M dataset, as they provide foreground-background segmentation masks. If the landmarks truly focus more on modeling the foreground more, then underlying activation heatmaps for each unsupervised landmark should be more contained within the provided segmentation masks in the factorized case. In Fig. 2b, we compare the percentage of the normalized activation maps contained within the provided segmentation masks against our baseline model for 8, 12, and 16 landmark models. For each learned landmark, we first compute its average activation mass contained within the foreground

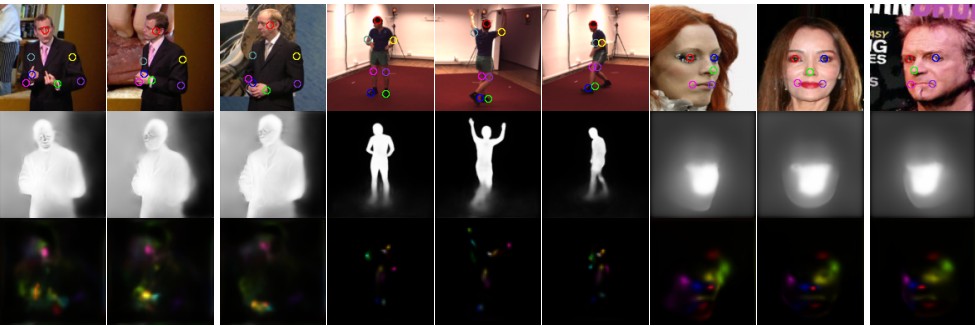

Figure 3: Qualitative results of our landmark prediction pipeline. From top to bottom, we show our regressed annotated keypoint predictions, our predicted foreground mask, and the underlying landmark activation heatmaps. Datasets are BBC Pose, Human3.6M, and CelebA/MAFL respectively.

segmentations. We then sort the landmarks in ascending order of containment (horizontal axis of Fig. 2b) and plot the models' landmark-containment curve.

The results in Fig. 2b demonstrate that the foreground-background factorization noticeably improves the least containment of the least-contained landmarks. Note that the lowest containment percentages for the baseline are 0.05, 0.4, and 0.3, whereas the factorized containment percentages are an order of magnitude larger at 8.2, 4.8, and 2.5 for 16, 12, and 8 landmarks respectively. It is safe to say that the least-contained landmarks for the baseline model are nearly completely utilized for modeling the image background (99%+ of the activation mass is on the background). While the proposed factorization does not eliminate the problem, we believe this difference is a contributing factor to the improvements over our baseline.

We show qualitative results of our regressed annotated keypoint predictions, as well as landmark activation and foreground mask visualizations in Fig. 3. From top to bottom, we show our regressed annotated keypoint predictions, our predicted foreground mask, and the underlying landmark activation heatmaps. Datasets are BBC Pose, Human3.6M, and CelebA/MAFL respectively. Notice that the degree of binarization in the predicted mask is indicative of the strength of the static background assumption on the data. Human3.6M features a strongly static background, whereas BBC Pose has a constantly updating display on the left, and CelebA was trained with $T_{tps}$ which indiscriminately warps both foreground and background. Nevertheless, our method still shows improves over the baseline despite imperfectly binarized foreground-background separation.

### 4.3 APPLICATION TO VIDEO PREDICTION

Lorenz et al. (2019) applied their model to video-to-video style transfer on videos of BBC signers, indicating that the rendered images from the landmark model are temporally stable. One of the issues with these renders, however, is that the landmarks are not suited for modeling the background, resulting in low-fidelity rendered backgrounds. We demonstrate that our factorized formulation resolves this issue.

We evaluate our rendering on the video prediction task on the KTH dataset, and compare against external methods. The unsupervised landmark model factorizes image data into pose (landmarks parameterized as 2D Gaussians) and appearance information. We assume the appearance information remains constant throughout each video sequence, and use an LSTM to predict how the 2D Gaussians move through time conditioned on an initial set of seed-frames. Refer to Appendix D for implementation details. We show our qualitative and quantitative results in Fig. 4 and 5 and respectively. We report SSIM, PSNR, and the perceptual-feature based LPIPS (Zhang et al., 2018a) metric. Note that the background-factorized approach significantly outperforms the unfactorized baseline on all performance metrics, indicating better background reconstruction, as the foreground is a comparatively smaller portion of the frame. Our method is also competitive with state-of-the-art models such as Lee et al. (2018). In Fig. 4, we show our rendered foreground, mask, rendered background, and the corresponding composition. Our method assumes a fixed background for the entire sequence, but predicts a new foreground and blending mask for each extrapolated timestep. Both our baseline and proposed method maintain better structural integrity than other methods. However, due to the imperfect binarization of the predicted mask, the foreground in the composite image may

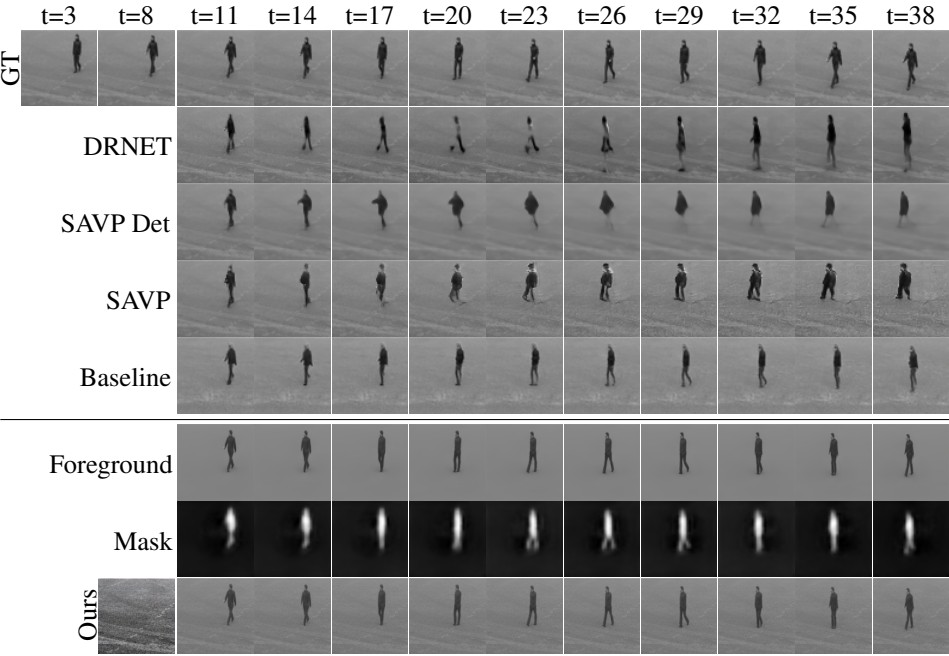

Figure 4: Qualitative results on KTH action test dataset comparing our method to prior work. Our baseline produces a sharp foreground, but the background does not match that of the initial frames. Our proposed factorized rendering significantly improves the background fidelity. The bottom three rows shows our factorized outputs. From top to bottom, we have the rendered foreground, the predicted blending mask, and the rendered background (first image on bottom row) followed by the composite output.

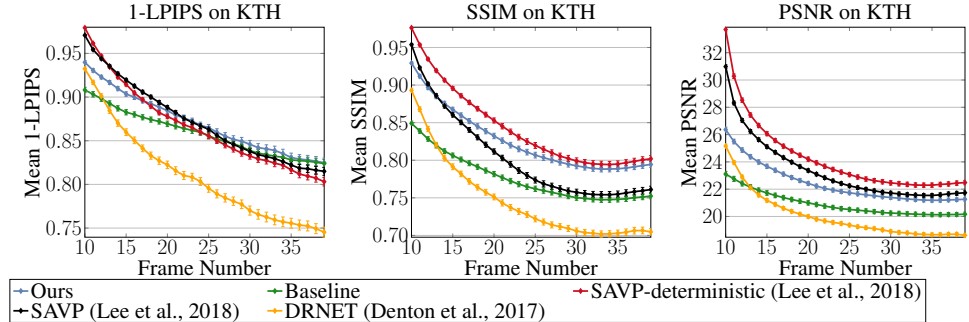

Figure 5: We base our main evaluation to LPIPS score which closely correlates with human perception. We also provide SSIM and PSNR metrics for completeness. Our method is competitive with state-of-the-art methods on KTH, and shows a large improvement over our controlled baseline.

appear somewhat faded compared to that of other methods. Improved binarization of the predicted masks remains a topic of future work.

## 5 CONCLUSION

We propose and study the effects of explicitly factorized foreground and background rendering on reconstruction-guided unsupervised landmark learning. Our experiments demonstrate that by using UNet to learn a simpler copy mechanism to copy roughly static background pixels, the model do a better job of allocating landmarks to the foreground objects of interest. As such, we are able to achieve more accurate regressions to annotated keypoints with fewer landmarks, thereby reducing memory requirements. We also demonstrate applications of our pipeline to unsupervised-landmark-based video manipulation tasks. For future work, we are interested in finding ways to improve binarization of the predicted foreground masks.

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

## A  IMPLEMENTATION DETAILS

**Architecture:** The overall architecture consists of 5 sub-networks as in: pose and appearance encoding networks, foreground mask subnet, background reconstruction subnet, and a foreground image decoder. We use the U-net architectures (Ronneberger et al., 2015) for the pose encoder, appearance encoder, foreground mask subnet and background reconstruction subnet, complete with skip connections. The pose encoder has 4 blocks of convolutional dowsampling modules. Each convolutional downsampling module has a convolution layer-Instance Normalization-ReLU and a downsampling layer. At each block, the number of filters doubles, starting from 64. The upsampling portion of the pose encoder has 3 blocks of convolutional upsampling modules, and the number of channels is halved at every block starting from 512. The appearance encoder network has one convolutional downsampling module and one convolutional upsampling module. The foreground mask subnet has 3 blocks of convolutional dowsampling module and 3 blocks of upsampling module, and the number of channels is 32 at each module. Similarly, the background reconstruction subnet has 3 blocks of convolutional dowsampling module and 3 blocks of upsampling module. At each block, the number of filters doubles starting from 32.

The image decoder has 4 convolution-ReLU-upsample modules. We first downsample the appearance featuremap by a factor of 8 in each spatial dimension. Number of output channels of each convolution-ReLU-upsampling module in the image decoder are 256, 256, 128, 64, and 3 respectively. We apply spectral normalization (Miyato et al., 2018) to each convolutional layer.

**Loss Function and Optimization Parameters:** We train our the image factorization-reconstruction network with VGG Perceptual loss which uses the pre-trained VGG19 model provided by the PyTorch library. We apply the MSE loss on outputs of layers `relu1_2`, `relu2_2`, `relu3_2`, and `relu4_2`, weighted by $\frac{1}{32}, \frac{1}{16}, \frac{1}{8}$, and $\frac{1}{4}$ respectively. We use Adam optimizer, learning rate of $1e^{-4}$, and weight decay of $5e^{-6}$. The network is trained on 8 GPUs with batch size of 16 images per GPU.

**Dataset Preprocessing:** For BBC Pose, we first roughly crop around each signer by using the given keypoints. Specifically, we find the center of the keypoints and crop $300 \times 300$ around the center and resize the crops to $128 \times 128$. For the Human3.6M dataset, we follow the procedure defined by Zhang et al. (2018b) for training/validation splits. We find the center of the keypoints and crop $300 \times 300$ around the center and again resize the crops to $128 \times 128$. For the CelebA/MAFL dataset, we follow Jakab et al. (2018) by resizing the images to $160 \times 160$, and center crop by $128 \times 128$.

Color jittering ($T_{cj}(x)$)is performed with `torchvision.transforms.ColorJitter(0.4, 0.4, 0.4, 0.3)` to the input image x.

The thin plate spline transformation $T_{tps}(x)$ allows us to perform a non-rigid warping of the image content based on applying perturbations to a grid of control points in the image's coordinate space. This was implemented with `cv2.createThinPlateSplineShapeTransformer()`. It is more expressive than a standard affine transformation on the image, allowing us to deform the image content in more interesting ways, and thus a reasonable drop-in replacement for $T_{temp}(x)$ when temporal data is not available.

**Regression coordinates** For BBC Pose and Human3.6M, we define the origin coordinate as the center of the image before regression. It is unclear what other methods used for these datasets. For CelebA/MAFL, we follow Jakab et al. (2018) and set the origin at the top left corner.

**Dataset-Specific Model parameters** We use 30 landmarks of fitted covariances for the BBC Pose dataset, meaning we estimate the covariance from the part activation maps when fitting the Gaussians to compute $\widetilde{\Phi}^{pose}$. For Human3.6M and CelebA, we use 16 and 10 landmarks respectively with a fixed diagonal covariance of 0.08. In general, fixed diagonal covariances lead to better performance on the landmark regression task than fitted covariance, though fitted covariances lead to better image generation results. As such, we use fitted covariances for the video prediction task.

## B  STATIC IMAGE REFACTORING DURING TRAINING

In Fig. 6, we show our model outputs from training on CelebA. We see that the pipeline has determined the center of the face to be foreground, with everything else as background. Importantly, the fourth column from the left shows that the $BGNet$ is capable of memorizing how to rectify a

| BBC Pose | Accuracy | |
|---|---|---|
| | Center | Top left |
| Baseline temp | 73.30 | 73.33 |
| Baseline (temp, tps) | 73.40 | 72.73 |
| Ours | 78.78 | 79.16 |

Table 3: BBC Pose evaluation with the origin of the coordinate space set at different locations. We reported center origin in the main paper. Note that the results differ despite coming from the same model output. Nevertheless, the improvement from our proposed method is still clear and the fluctuation in accuracy remains small.

thin-plate-spline warped image. While this is a case of overfitting, it is arguably the reason our approach is able to perform foreground-background separation when the static background assumption is weak. A median-filtering based approach for background subtraction would be feasible only on video sequence data with perfectly still backgrounds.

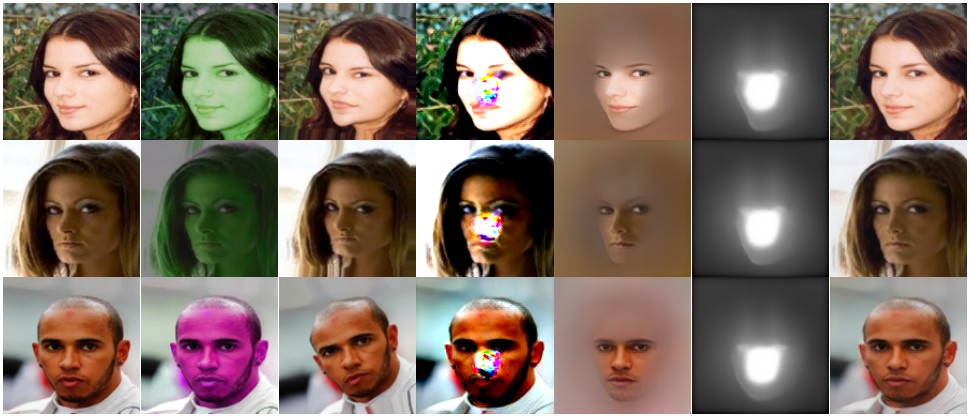

Figure 6: From left to right input image, color jittered image, thin-plate-spline warped image, reconstructed background, predicted foreground, mask, and reconstructed output.

## C  ADDITIONAL QUALITATIVE LANDMARK PREDICTION RESULTS

Here, we show additional results for the pose-regression task on various datasets. The regression quality is generally very accurate, though notice that, as with other unsupervised landmark approaches, we cannot model keypoint visibility easily, nor can we distinguish between front and back facing subjects on the Human3.6M dataset.

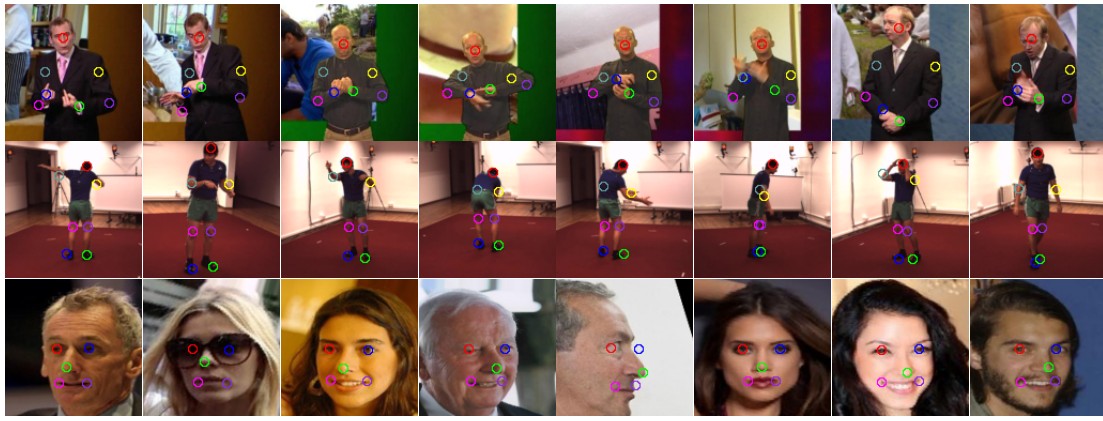

Figure 7: Additional qualitative results for keypoint predictions for BBC Pose, Human3.6M, and CelebA/MAFL respectively.

# D    VIDEO PREDICTION WITH UNSUPERVISED LANDMARKS

As can be seen in Fig. 1, the rendering component of our pipeline is conditioned directly on an appearance encoding $\Phi^{app}$ and a pose encoding $\widetilde{\Phi}^{pose}$. Conveniently, the pose representation is stored as a set of 2D Gaussian distributions, which can can easily manipulate spatially by translating their means and updating their covariances. As such, we apply our model to the video prediction task by assuming $\Phi^{app}$ to be constant throughout the prediction sequence, and using an LSTM to update $\widetilde{\Phi}^{pose}$, conditioned on the first $N$ frames. We extract $\Phi^{app}$ from the $N^{th}$ frame in the input sequence.

Care must be taken to maintain positive definite covariance matrices during prediction. Thus we use the parameters of the Cholesky decomposition of the covariance matrices as the prediction targets. In practice, the LSTM may produce an estimate for $L$ which is not a valid Cholesky factor, but even this case will still produce a valid covariance matrix when $LL^T$ is computed.

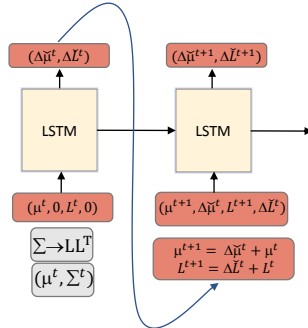

An illustration of our LSTM setup is depicted in Fig.8. In addition to the Cholesky parameterization, we also found that predicting the residual during extrapolation, rather than the state directly, was important to robust long term predictions. In practice. this helped improve both training performance and generalization. At each timestep, the LSTM takes a concatenation of the previous state and state residual to predict the next state residual.

Figure 8: The LSTM predicts perturbations to the Gaussian means and Cholesky parameters of the covariances.

Our LSTM comprises 3 LSTM layers and a final linear layer. Each LSTM layer has 256 channels. For the KTH results, we trained our landmark model with 40 landmarks. We also use a GAN loss term to improve image quality. Due to the KTH dataset being grayscale, we had to predict foreground-background masks at half resolution to prevent the masks from encoding all the foreground details. The LSTM was trained to predict 10 future frames from an input sequence of 10.

## E  ADDITIONAL VIDEO PREDICTION RESULTS

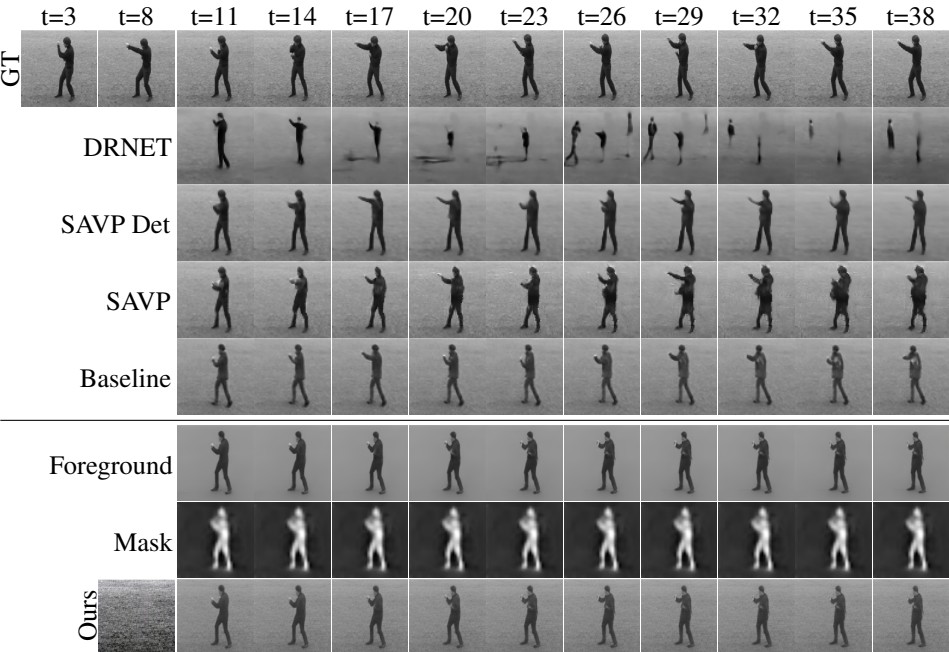

Figure 9: Qualitative results on KTH action test dataset comparing our method to prior work. SAVP and SAVP-deterministic methods produce blurry foreground images. On the other hand, the baseline method produces sharp foreground but the background does not match the initial frames. Our method maintains sharpness and high fidelity to the background. We show the foreground image reconstructions and masks that are used to produce output images. In the last row, first column shows the reconstructed background image.

## F  ADDITIONAL VIDEO PREDICTION EXPERIMENT ON BAIR PUSH DATASET

We additionally run video prediction experiments on the BAIR action-free dataset (Finn et al., 2016). This dataset consists of videos with robot arms moving randomly with a diverse set of objects on a table. The videos have spatial resolution of 64×64. For this dataset, our video prediction LSTM is trained with a 10 input 0 future setup (never conditioning on its own output during training). We show our qualitative and quantitative results on the BAIR dataset in Fig. 10 and 11 and respectively.

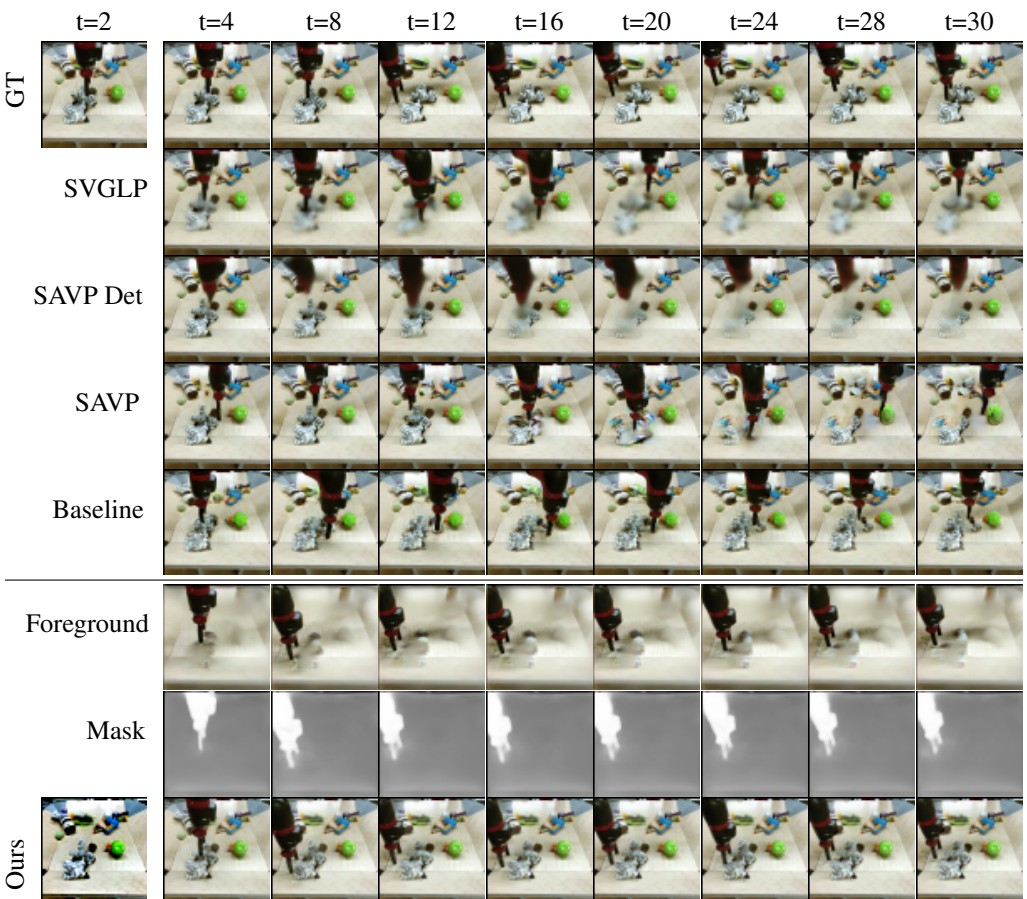

Figure 10: Qualitative results on the BAIR dataset comparing our method to prior work. Methods are conditioned on 2 initial frames to predict the next 28. SVGLP, SAVP and SAVP-deterministic methods produce blurry outputs in the previously occluded regions. Our method maintains sharpness and high fidelity to the background. We show the foreground image reconstructions and masks that are used to produce output images. In the last row, first column shows the reconstructed background image.

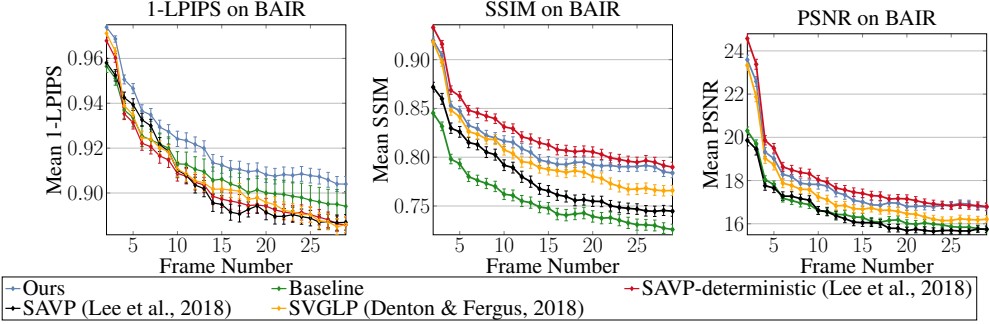

Figure 11: We base our main evaluation to LPIPS score (Zhang et al., 2018a) which closely correlates with human perception. We also provide SSIM and PSNR metrics for completeness. Our implementation achieves better LPIPS score than the competing methods. Importantly, the factorized method significantly outperforms our baseline by a large margin on all metrics.

