# OpenReview forum: "Unsupervised Disentanglement of Pose, Appearance and Background from Images and Videos"
_ICLR.cc/2020/Conference — Reject_

### Official Review · AnonReviewer1 · 2019-10-19
**Official Blind Review #1**

**Rating:** 3

**Review:**

In this paper, authors propose to design an unsupervised learning framework, which can capture pose representation by reconstructing images or videos.

The paper is not written well. Too much components in the design make this work hard to follow. The novelty is also relatively limited, since a large part of this work has been done in Lorenz et al. (2019). Moreover, the only supervision is the reconstruction loss. I am just curious how the neural network can learn the semantics representation (such as foreground and background), without any guidance in the corresponding modules?


**Experience Assessment:**

I have read many papers in this area.

**Review Assessment: Checking Correctness Of Derivations And Theory:**

I assessed the sensibility of the derivations and theory.

**Review Assessment: Checking Correctness Of Experiments:**

I assessed the sensibility of the experiments.

**Review Assessment: Thoroughness In Paper Reading:**

I made a quick assessment of this paper.

---

> ### Author Response · Authors · 2019-11-11
> **Clarity, better intuition**
>
> Thank you for your feedback. We have provided additional explanation in both the global comment and below to better explain the contributions and mechanisms of our work.
>
> Unsupervised landmarks are desirable as they allow us to perform tasks that may require keypoint-level annotations. Ideally, we want these landmarks to focus on the foreground, but this is difficult to guarantee. Our work demonstrates that if one can roughly adhere to a static background/moving-foreground assumption (please see the global comment) within the training data, a straightforward factorization of the reconstruction pipeline will result in more foreground-focused landmarks.
>
> Another key contributions is our detailed experimental analysis, most notably the results in Figure 2, in which we manage to quantitatively confirm that the landmarks are, in fact, more focused on the foreground when the assumptions hold. Furthermore, we demonstrate that having landmarks more focused on the foreground means we can train landmarks models with fewer landmarks, as each additional landmark we use significantly increases the model resource requirement.
>
> We hope that our additional explanations provide enough intuition as to how and why the foreground and background are separated in our approach. This information is mentioned in our experimental analysis section, but we have updated our draft to include this intuition early on in the methods section for better clarity.

---

### Official Review · AnonReviewer2 · 2019-10-22
**Official Blind Review #2**

**Rating:** 6

**Review:**

The paper presents an unsupervised approach for learning landmarks in images or videos with single objects by separating the representation of the image into foreground and background and factorizing the representation of the foreground into pose and appearance. It builds upon previous work [Jakab 2018, Lorenz 2019] who proposed to train by reconstructing the original image from one version of the image with perturbed appearance and another with perturbed pose. It extends this approach by introducing an additional separation of foreground and background in the image.

Strengths:
+ Nicely motivates the approach of separating foreground and background
+ Fewer landmarks are needed than in previous work
+ Approach seems beneficial for video prediction
+ Clear and well written
+ Detailed description of architecture and training

Weaknesses:
- The changes and improvements feel somewhat incremental
- Some uncertainty about the solidity of the evaluation/comparability with baselines
Results on CelebA somewhat weak

Overall the paper is well written, easy to follow, presents a straightforward extension of previous work and appears to show an improvement. I’m thus generally supportive of the paper.

One question I’d like to see addressed in the response is about the evaluation: as you state, the details about cropping are not known for previous work, introducing some uncertainty into the external comparisons. It would be great if you could provide some more details about the steps you took to verify that your internal baseline is indeed comparable to previous work (e.g. Lorenz 2019). For instance, on CelebA your baseline seems to fall short of Lorenz 2019, which may suggest that your substantial looking improvement on BBC Pose is indeed due to more favorable cropping.

The second concern is about the results on CelebA presented in Fig. 6 in the Appendix: It looks like the background net reconstructs almost the entire image. I would have expected that hair style, shape of ears or existence of a beard would equally warrant landmarks and appearance. It seems a bit odd that the foreground is so focused on the central part of the face. Do you have an explanation for that?

Minor comments:
- Test in Fig. 1 too small and not readable when printed
- Fig. 1 seems to be missing an arrow pointing from image to appearance encoder
- Why are there more than 8 points in Fig. 2 for “Ours8” (and more than 12 for “Ours12”)?


**Experience Assessment:**

I have read many papers in this area.

**Review Assessment: Checking Correctness Of Derivations And Theory:**

I assessed the sensibility of the derivations and theory.

**Review Assessment: Checking Correctness Of Experiments:**

I assessed the sensibility of the experiments.

**Review Assessment: Thoroughness In Paper Reading:**

I read the paper thoroughly.

---

> ### Author Response · Authors · 2019-11-11
> **Evaluation protocol reliability, better intuition, fixed figures**
>
> Thank you for your detailed review and constructive comments.
>
> Matching the performance of Lorenz et al.:
> We would first like to note that the primary comparison should be made against our internal baselines, as only in that case are the base architectures, losses, and data augmentations fully controlled.
>
> Here, we detail some of the steps we took to ensure that our baseline architecture performs closely to that of Lorenz et al. Our reconstruction losses and generator architecture differ, but we were able to confirm fairly close (slightly worse) landmark regression performance to the best of our ability. Specifically, on BBC-Pose, we matched their cropping parameters exactly, which we received from the authors via email correspondence. For the regression coordinate space, we tested two variants: (1) the origin is in the center of the crop; (2) the origin is the top left corner of the crop. For each case, the origin placement rule is applied to both the landmarks and the target annotated keypoints. We include the results in Table 3 of the appendix of the now updated draft. The keypoint accuracy varies only a little between the two cases, and the improvement from incorporating the background factorization remains large. From this, we can reasonably conclude that our approach significantly outperforms Lorenz et al, whereas our baseline performs about 1% worse than Lorenz et al. on BBC Pose (and similarly slightly worse on CelebA as noted). We are reasonably confident that our improvement from background factorization can be easily incorporated into the exact architecture used by Lorenz et al. as well.
>
> Poor foreground background separation on CelebA:
> Firstly, as mentioned in our main comment, CelebA was meant as a stress test dataset in which the static background assumption is the weakest. We did not expect a clean separation of the foreground and background on this dataset, though it was interesting to see that we were still able to improve over our baseline.
>
> As for why the BGNet reconstructs the entire image, the most likely explanation is that the foreground and background rendering tasks were never cleanly factorized on this dataset, given the extremely weak static background/moving foreground assumptions inherent to our approach. As such, both the foreground rendering and background rendering components attempt to reconstruct the face — the former via conditioning on the placement of landmarks, and the latter based on unwarping the thin-plate-spline-warped image.
>
> The heavy focus on the central part of the face can be explained by its low level of visual variance. Landmarks are meant to generalize across all images in the dataset. This is easiest to achieve if their underlying “part” has very low visual variance, such as in the case of noses, eyes, and mouths. The shape, color, and texture of the hair can vary significantly from person to person — almost as much as the background does from image to image in this dataset. Similarly, the ears may or may not even be visible, depending on whether it is occluded by the hair. In the absence of the BGNet, the model would be forced to use landmarks to model every visual component within the image. With the BGNet, the mask weights suggest that learning a set of landmarks that can consistently localize the hair and ears across all images is much harder than treating that part of the image as background and learning to undo the TPS warping.
>
>
> Fixed errors in figures:
> Thank you for catching these!
>
> We increased the font size of the text in Figure 1. The appearance encoder actually shares the first conv layer with the shape encoder as was the case in Lorenz et al. There is an arrow pointing from the first layer to the appearance encoder right now.
>
> We have also fixed the plotting in Figure 2(b), which was apparently replicating the last data point to extend each curve up to 16 landmarks.
>
> The draft has been updated to reflect these changes.

---

### Official Review · AnonReviewer3 · 2019-10-23
**Official Blind Review #3**

**Rating:** 6

**Review:**

The paper presents an unsupervised method to get disentanglement of pose, appearance, background from both images domain and video domain. 5 sub-network are used to model pose, appearance, foreground, background, and decoders.  Their methods let the network focus more on the foreground to regress the landmark and improve state-of-the-art performance on landmark regression (unsupervised.), video prediction and image reconstruction.
However, there are still lots of details missing for the training of the whole network even with the supplementary.
1. what are the details of the color jitter process? how do you know it is foreground and only colorizing this part?
2. why the video prediction only on KTH. H36M is also a video-based dataset.
3. what is the thin-plate-spline warped image?
4. how do you generate T_temp(x) for image-based dataset?
5. are the learned landmark all unimodal? as for 2d pose estimation, even we give them unimodal gt, sometimes the prediction is bimodal.
6. how do you make the covariance as learned?

**Experience Assessment:**

I have published one or two papers in this area.

**Review Assessment: Checking Correctness Of Derivations And Theory:**

I carefully checked the derivations and theory.

**Review Assessment: Checking Correctness Of Experiments:**

I assessed the sensibility of the experiments.

**Review Assessment: Thoroughness In Paper Reading:**

I read the paper at least twice and used my best judgement in assessing the paper.

---

> ### Author Response · Authors · 2019-11-11
> **Clarifications and implementation details**
>
> Thank you for pointing out the missing implementation details. We have updated our implementation details and hope our responses below sufficiently address your concerns.
>
> Color jitter process:
> To recap the method, the model accepts two variants of the image — $T_{cj}(x)$ and $T_{temp}(x)$, the former of which contains the correct pose, but the perturbed appearance (mostly incorrect colors), and the latter of which contains the incorrect pose but correct appearance. $T_{cj}(x)$ is implemented by applying torchvision.transforms.ColorJitter(0.4, 0.4, 0.4, 0.3) to the input image x. We do not need to limit the color jittering to just the foreground, as we satisfy our goal by ensuring that none of the appearance/color information anywhere in $T_{cj}(x)$ is usable.
>
>
> Why not H36M for video prediction:
> The papers that we chose to compare against used KTH (in the main paper) as well as BAIR push (in the supplemental - Appendix F) datasets. We are certainly aware of other works that test on H36M which we could have compared to as well. However, the main purpose of the video prediction results was to demonstrate improved background reproduction fidelity compared to our controlled baseline. For this, we considered our experimentation to be sufficient and did not try H36M due to time and resource constraints.
>
> Thin-plate spline warped image?/How to generate T_{temp}(x) for image-based datasets:
> The thin plate spline transformation allows us to perform a non-rigid warping of the image content based on applying perturbations to a grid of control points in the image’s coordinate space.  This was implemented with cv2.createThinPlateSplineShapeTransformer(). It is more expressive than a standard affine transformation on the image, allowing us to deform the image content in more interesting ways, and thus a reasonable drop-in replacement for $T_{temp}(x)$ when temporal data is not available. We cannot use $T_{temp}$ for image-based datasets, and thus are forced to rely on $T_{tps}$ as a reasonable alternative. The CelebA results were meant to test the limits of our method where the static background and moving foreground assumptions broke down. Unsurprisingly, the results were not as clean a separation of foreground background as in our video based datasets, yet we still saw a marked improvement in landmark quality.
>
> Unimodality of landmarks:
> This is a fantastic question! Yes, as with prior architectures, we fit a single 2D gaussian on top of each part activation heatmap, which absolutely assumes a unimodal underlying heatmap during both training and test. One would hope that the final converged model has a decent set of landmarks for which the underlying distribution is also unimodal, but that cannot be guaranteed. This is a fundamental limitation of current methods, and would be an interesting direction for future work.
>
>
> Covariance:
> We have two types of models, a fixed covariance model, where we assume 2D isotropic gaussian landmarks with a fixed covariance matrix (compute only the mean of activation map), and a fitted covariance model (compute both mean and covariance of the activation map). The covariance matrix parameters are never directly predicted by the landmark model.

---

### Author Response · Authors · 2019-11-11
**Global Comment**

Reviewers:
Thank you for your valuable feedback. Feedback was mostly positive, but clarifications and some additional experiments were requested.
R1 and R2 both asked for additional intuition and explanations of our proposed method’s behavior
R3 requested additional implementation details
R2 requested additional verification that our improvement over prior works is not largely due to differences in unspecified details in evaluation protocols (cropping, coordinate space definitions).

Here, we first address the intuition requested from R1 and R2. Remaining points will be addressed in direct responses.


R1 and R2: Intuition on why the foreground-background separation happens:
During training the model can reconstruct the image using by copying from the pos-perturbed input or by performing conditional image generation using the latent landmark representation. Controlling the relative ease with which each approach can be learned to reconstruct the input is the key to our approach’s success. The main intuition is that if we assume a moving foreground and static background, then the pose change in $T_{temp}(x)$ will correspond mostly to the moving foreground, and that the majority of the background can be copied to the reconstruction with a set of skip connections. As such, we expect the BGNet to learn this simple copy mechanic quickly during training, as the BGNet architecture is a simpler feedforward UNet with skip connections. What remains is the foreground that has moved, which requires the landmarks to find part-level correspondences between $T_{cj}(x)$ (same pose as reconstruction target) and $T_{temp}(x)$.

The difficulty of each dataset with respect to our method is related to how strongly the static-background/moving foreground assumption holds. H36M is the easiest because the background is completely static throughout any given sequence. BBC is roughly static, with a dynamic background to the left of the presenter. CelebA is the extreme case we must use thin-plate-spline warping in lieu of temporal video sampling. This makes the job of the BGNet much harder as thin-plate-spline warping is indiscriminately warps both the foreground and background content. This ranking of dataset difficulty is empirically reflected in the degree of mask binarization in Figure 3 of the paper, with H36M being the most binarized/separable. Importantly, our method still yields improvements in landmark quality when the assumptions hold weakly.

---

### Decision · Program_Chairs · 2019-12-19

**Decision:**

Reject

**Comment:**

The paper proposes an approach for unsupervised learning of keypoint landmarks from images and videos by decomposing them into the foreground and static background. The technical approach builds upon related prior works such as Lorenz et al. 2019 and Jakab et al. 2018 by extending them with foreground/background separation. The proposed method works well for static background achieving strong pose prediction results. The weaknesses of the paper are that (1) the proposed method is a fairly reasonable but incremental extension of existing techniques; (2) it relies on a strong assumption on the property of static backgrounds; (3) video prediction results are of limited significance and scope. In particular, the proposed method may work for simple data like KTH but is very limited for modeling videos as it is not well-suited to handle moving backgrounds, interactions between objects (e.g., robot arm in the foreground and objects in the background), and stochasticity.